# Chalcones identify cTXNPx as a potential antileishmanial drug target

Douglas O. Escrivani[1,2], Rebecca L. Charlton[1,2], Marjolly B. Caruso[3], Gabriela A. Burle-Caldas[4], Maria Paula G. Borsodi[1], Russolina B. Zingali[3], Natalia Arruda-Costa[1], Marcos V. Palmeira-Mello[5], Jéssica B. de Jesus[5], Alessandra M. T. Souza[5], Bárbara Abrahim-Vieira[5], Stefanie Freitag-Pohl[2], Ehmke Pohl[2,4], Paul W. Denny[4], Bartira Rossi-Bergmann[1]*, Patrick G. Steel[2]*

1 Instituto de Biofísica Carlos Chagas Filho, Universidade Federal do Rio de Janeiro, Rio de Janeiro, Brazil, 2 Department of Chemistry, Durham University, Science Laboratories, South Road, Durham, United Kingdom, 3 Instituto de Bioquímica Médica Leopoldo de Meis (IBqM), Universidade Federal do Rio de Janeiro, Rio de Janeiro, Brazil, 4 Department of Biosciences, Durham University, Science Laboratories, South Road, Durham, United Kingdom, 5 Faculdade de Farmácia, Universidade Federal do Rio de Janeiro, Rio de Janeiro, Brazil

* bartira@biof.ufrj.br (BR-B); p.g.steel@durham.ac.uk (PGS)

**Data Availability Statement:** All relevant data are within the manuscript and its Supporting Information files.

## Abstract

With current drug treatments failing due to toxicity, low efficacy and resistance; leishmaniasis is a major global health challenge that desperately needs new validated drug targets. Inspired by activity of the natural chalcone 2',6'-dihydroxy-4'-methoxychalcone (DMC), the nitro-analogue, 3-nitro-2',4',6'-trimethoxychalcone (NAT22, **1c**) was identified as potent broad spectrum antileishmanial drug lead. Structural modification provided an alkyne containing chemical probe that labelled a protein within the parasite that was confirmed as cytosolic tryparedoxin peroxidase (cTXNPx). Crucially, labelling is observed in both promastigote and intramacrophage amastigote life forms, with no evidence of host macrophage toxicity. Incubation of the chalcone in the parasite leads to ROS accumulation and parasite death. Deletion of cTXNPx, by CRISPR-Cas9, dramatically impacts upon the parasite phenotype and reduces the antileishmanial activity of the chalcone analogue. Molecular docking studies with a homology model of *in-silico* cTXNPx suggest that the chalcone is able to bind in the putative active site hindering access to the crucial cysteine residue. Collectively, this work identifies cTXNPx as an important target for antileishmanial chalcones.

## Author summary

Leishmaniasis is an insect vector-borne parasitic disease. With >350 million people world wide considered at risk, 12 million people currently infected and an economic cost that can be estimated in terms of >3.3 million working life years lost, leishmaniasis is a major global health challenge. The disease is of particular importance in Brazil. Current treatment of leishmaniasis is difficult requiring a long, costly course of drug treatment using old drugs with poor safety indications requiring close medical supervision. Moreover, resistance to current antileishmanials is growing, emphasising a major need for new drug

**Funding:** We thank the Royal Society (International Collaboration Award to Research Professors to PGS & BRB, ICA-IC1600) https://royalsociety.org; the UKRI - Global Challenges Research Fund. 'A Global Network for Neglected Tropical Diseases' MR/P027989/1 (to PWD, EP, PGS, BRB) https://www.ukri.org; Conselho Nacional de Pesquisa Tecnológica -Brasil (CNPq) 400894/2014-9 (to BRB and PGS) http://www.cnpq.brand the Coordenação de Aperfeiçoamento de Pessoal de Nível Superior – Brasil (CAPES) (to BRB) https://www.gov.br/capes/pt-br for financial support of this work. The funders had no role in study design, data collection and analysis, decision to publish, or preparation of the manuscript.

**Competing interests:** The authors have declared that no competing interests exist.

targets. In earlier work we had identified a naturally inspired chalcone which had promising antileishmanial activity but with no known mode of action. In this work we use an analogue of this molecule as an activity based probe to identify a protein target of the chalcone. This protein, cTXNPx, has a major role in protecting the parasite against attack by reactive oxygen species in the host cell. By inhibiting this protein the parasite can no longer survive in the host. Collectively this work validates cTXNPx as a drug target with the chalcone as a lead structure for future drug discovery programmes.

## Introduction

The leishmaniases are a complex group of vector-borne diseases caused by protozoan parasites of genus *Leishmania*. Endemic in over 98 countries, placing in excess of 1 billion people at risk of infection [1, 2] and with an economic burden best represented by more than 2.4 million DALYs (disability-adjusted life years), leishmaniasis is considered by the WHO as one of the most important neglected tropical diseases [3]. Clinical manifestations vary according to the parasite species and the host immunological status [4]. These range from the more benign skin ulcers in cutaneous leishmaniasis (CL) to visceral leishmaniasis (VL), which infects the spleen, liver, and bone marrow and it is fatal in more than 95% of untreated cases [4]. With no human vaccine available, treatment relies on chemotherapy and the current pharmacopeia is reliant on a limited number of drugs: pentavalent antimonials, amphotericin B, pentamidine, miltefosine and paramomycin [5]. All of these possess drawbacks, including poorly-defined modes of action, invasive routes of administration, severe side effects, variable efficacy, high cost and growing resistance [6]. With approximately 1 million new cases of CL and 90,000 cases of VL each year worldwide, there is a major unmet need to develop new broad-spectrum chemotherapies based on new validated drug targets [7].

Chalcones have been described as a potential new class of antileishmanials. More than 300 chalcones have reported activity against *Leishmania* parasites [8–10]. We have previously reported the selective antileishmanial activity of a natural chalcone 2',6'-dihydroxy-4'-methoxychalcone (DMC) **1a** (Fig 1) extracted from the pepper plant *Piper aduncum* [11]. Subsequent synthetic refinement afforded the nitro-chalcone, 3-nitro-2'-hydroxy-4',6'-dimethoxychalcone (CH8) **1b** [12], which showed strong and selective anti-parasite activity *in vitro* and *in vivo* against both *Leishmania amazonensis* and *Leishmania infantum* [13–16]. Despite those promising results, further enhancement to activity were challenged by the fact the mechanism of action of CH8 **1b** remained to be elucidated. In this report, through a combination of chemical, molecular and cellular biology, we describe the use of a CH8-derived chemical probe **1c** (3-nitro-2',4',6'- trimethoxychalcone, NAT22) to identify the cytosolic enzyme tryparedoxin peroxidase (cTXNPx) as a target for chalcones in a range of *Leishmania* species. Additionally, we demonstrated that the interaction between the chalcone and cTNXPx inhibits the capacity to detoxify reactive oxygen species (ROS) leading to parasite death. As such, this study demonstrates that this parasitic specific protein has significant potential as a drug target for the future treatment of leishmaniasis.

## Experimental

### Analogues and probe synthesis

Details concerning the chemical synthesis and compound characterization are described in the Supplementary Information (S1 Text).

**Fig 1. Chemical structures of compounds 1–4.**

## Anti-promastigote activity

Promastigotes of *L. amazonensis* (strain MHOM/BR/75/Josefa, 5 x $10^5$/mL), *L. braziliensis* (strain MHOM/BR/75/M2903, 2 x $10^6$/mL) and *L. infantum* (strain MHOM/MA67IT-MAP263, 2 x $10^6$/mL) at the logarithmic phase of growth, were incubated at 26˚C with different concentrations of compounds **1c**, **2** and **3** for 72 h in M199 medium containing 5% of heat inactivated fetal bovine serum (HIFBS), 0.2% hemin (Sigma-Aldrich), 100 µg/mL streptomycin and 100 UI/mL penicillin (Sigma-Aldrich). Parasite viability was assessed in the last four hours of culture by Alamar blue (ThermoFisher Scientific). The concentrations that reduced cell viability by 50% ($EC_{50}$) were calculated by non-linear regression analysis using GraphPad Prism 7 software.

For mutant cell lines, promastigotes of *L. mexicana* T7 Cas9 (Parental), Δ1040, Δ1160 and ΔcTXNPx (2 x $10^6$/mL), were incubated at 26˚C with different concentrations of compounds **1c** for 48 h in Schneider's medium (Sigma-Aldrich) containing 10% of heat inactivated fetal bovine serum (HIFBS), 100 µg/mL streptomycin and 100 UI/mL penicillin. Parasite viability and $EC_{50}$ values were assessed as described above.

## Anti-amastigote activity

Bone marrow-derived macrophages (BMDM) were differentiated from BALB/c mouse bone marrow [17] and plated at 5 x $10^5$/well in 24-well culture plates onto circular glass coverslips in RPMI medium containing 10% of HIFBS at 37˚C/5% $CO_2$ for 24 h. Then the macrophage monolayers were infected with *L. amazonensis* promastigotes (5 x $10^6$/well) at 34˚C for 4 h. After removal of free parasites by washing with phosphate buffered saline (PBS), the cells were

cultured at 37˚C for further 24 h for parasite differentiation into amastigote forms. Infected cells were treated with different concentrations of compounds **1c, 2** or **3** for 48 h, washed with PBS and stained with Panotic stain (Newprov, Brazil) according to manufacturer instructions. At least 200 macrophages/coverslip and their intracellular amastigotes were counted at 400x magnification for the determination of $EC_{50}$ values as described above.

## Macrophage cytotoxicity

Uninfected BMDM were treated as above for infected cells. At the end of the 48 h incubation time, the specific release of the cytoplasmic enzyme lactate dehydrogenase (LDH) in culture supernatants was measured using a commercial assay kit (Doles, Brazil), as previously described [15]. $CC_{50}$ values were calculated as described for $EC_{50}$ values. The selectivity index (SI) for each compound was calculated as the ratio between macrophage cytotoxicity ($CC_{50}$) and anti-amastigote activity ($EC_{50}$).

## Mouse infection and oral treatment with chalcone 1c

All animal studies were approved by the Animal Care and Use Committee of the Health Sciences Center of the Federal University of Rio de Janeiro, Brazil, under the protocol number CEUA 030/17. All animal experiments were conducted in compliance with the principles stated in the *Guide for the Care and Use of Laboratory Animals, 8th Edition.*[18] BALB/c mice were randomly distributed into groups of 5 and intradermally infected in the right ear pinna with $2 \times 10^6$ *L. amazonensis* promastigotes at stationary phase of growth [16]. One week after infection, animals were orally treated daily for 3 weeks by intragastric gavage with compound **1c** (40 mg/Kg/day) in propylene glycol (PPG) vehicle. Untreated controls received PPG vehicle alone. For clinical follow-up, ear thicknesses were periodically measured using a calliper gauge. For parasite burden, on day 30 of infection, the individual ear tissue homogenates were assayed by limiting dilution (LDA) [15]. Animal sample size was estimated by a power-based method using the following assumptions: $\alpha = 0.05, 1 - \beta = 0.9$, and standard deviation 20% of mean for two groups [19].

## Intracellular chalcone distribution by confocal microscopy

*L. amazonensis* promastigotes ($1 \times 10^6$/mL) were incubated with compound **2** (5 μM) at 26˚C in 1 mL of medium M199 for 5 h at 26˚C. At the end of incubation time cells were washed 3 times with PBS, fixed with 4% of paraformaldehyde and permeabilized with 0.1% saponin in PBS for 5 min at room temperature (RT). Fixed/permeabilized parasites were then subjected to cycloaddition reaction (CuAAC) by incubation with fluorescent 'trifunctional' probe **4** (Fig 1; 5 μM) in the presence of $CuSO_4$ (1 mM), sodium ascorbate (2 mM) and tris-hydroxypropyl-triazolylmethylamine (THPTA, 2 mM) for 1h at room temperature. Then, cells were incubated for 15 min with Hoechst 33342 (8 μM) for nucleus and kinetoplast labelling. Cells were washed with PBS, placed onto microscope slides and imaged using confocal microscope (Leica TCS-SPE) at 63x (ACS APO 63x 1.30 oil objective) in the Hoechst (350 / 461 nm), TAMRA (555 / 580 nm) wavelengths and DIC (Differential interference contrast).

## Intracellular protein tagging by SDS-PAGE and Western blotting

*L. amazonensis* promastigotes ($6 \times 10^7$/mL), BMDM ($2 \times 10^6$/well) and amastigotes isolated from infected macrophages [20], were treated with compound **2** (1 μM and 5 μM) for 4 h at 27˚C (promastigotes) or 37˚C (BMDM and amastigotes). Then, cells were washed twice with cold PBS, lysed by sonication in lysis buffer (1% sodium deoxycholate, 0.5% SDS, 50 mM Tris pH 7.4, 150 mM NaCl) containing EDTA-free protease inhibitor cocktail (Sigma-Aldrich).

Proteins were precipitated with acetone (4 volumes at −20˚C for 1 h), and quantified using Bicinchoninic Acid Kit (BCA1, Sigma Aldrich) according to manufacturer's protocol.

For competition assay, promastigotes ($6 \times 10^7$/mL) were incubated with **1c** at different concentrations (1, 3, 9, 27 μM) for 4 h, washed twice with PBS, followed by incubation with **2** (1 μM) for additional 4 h in M199 medium at 26˚C. Cells were then lysed, and parasite proteins precipitated and quantified as above.

For unidimensional electrophoresis (SDS-PAGE), proteins (1 mg/mL) were incubated with compound **4** (5 μM), CuSO$_4$ (1 mM), sodium ascorbate (2 mM) and THPTA (2 mM). The click reaction was carried out for 1 hr at RT, then quenched by addition of 4x SDS-PAGE sample buffer with 50 mM of DTT (Bio-Rad). Proteins were then heated for 5 min at 95˚C, run in 10% SDS-PAGE, and gel images taken at Gel Doc XR imager (Bio-Rad) using Comassie Blue and Oriole filter (270 / 604 nm) for compound **4** fluorescence detection. 2D electrophoresis analyses were performed as described by Higa et al [21].

## Protein identification by mass spectrometry

Proteins from promastigotes treated with **2** and linked to **4** were separated by unidimensional SDS-PAGE as above. The gel section of interest (~22 kDa) was manually excised and incubated overnight with 25 mM NH$_4$HCO$_3$ in 50% acetonitrile to remove the stain. Then proteins were reduced for 1h at 56˚C with 10 mM DTT (USB Corporation), alkylated for 45 min at room temperature with 55 mM iodoacetamide and digested by trypsin (modified sequencing grade; Promega) solution (0.01 μg/μL in 25 mM NH$_4$HCO$_3$) overnight at 37˚C. Generated peptides were extracted with 5% trifluoroacetic acid (TFA) and 50% acetonitrile, followed by concentration in a Speed-Vac system. Each sample was then solubilized in 20 μL 0.1% formic acid and 3% acetonitrile in deionized water. The mass spectrometry analyses were performed as described previously [22]. In brief, the peptides were first desalted on-line using a Waters Symmetry C18 trap column, loaded on a Waters Nanoacquity UPLC system (Waters, Milford, MA) and finally sprayed into a Q-Tof quadrupole/orthogonal acceleration time-of-flight spectrometer (Waters, Milford, MA) interfaced with the Nanoacquity system capillary chromatograph. The exact mass was determined automatically using the Q-Tof LockSpray (Waters, Milford, MA). Data-dependent MS/MS acquisitions were performed on precursors with charge states of 2, 3 or 4 over a range of 50–2000 *m/z* and under a 2 *m/z* window.

## Production and purification of recombinant cTXNPx

Expression plasmid pPOPINF containing ORF for cTXNPx from *L. amazonensis* (AY842247.1) tagged with N-terminal 6xHIS-tag was transformed into BL21 (DE3) competent cells (Agilent Technologies). One litre LB-ampicillin cultures were induced to express recombinant protein by addition of 1M isopropyl β-D-1-thiogalactopyranoside (IPTG). After 20 h of incubation at 30˚C, cells were pelleted and resuspended in 50 mL of lysis buffer (20 mM Tris-HCl pH 7.5, 500 mM NaCl, 30 mM imidazole and 2 mM β-mercaptoethanol) with one tablet of Sigma-Fast protease inhibitors (Sigma-Aldrich). Cell pellets were further lysed by sonication and proteins were purified using His-tag affinity chromatography (HisTrap HP-1 mL column, GE healthcare). The cTXNPx peak fractions were pooled and concentrated to approximately 1 mg/mL as determined spectrophotometrically using a NanoDrop at 280 nm.

## Chalcone and recombinant cTXNPx interaction by SDS-PAGE, LC-MS/MS and TSA

For SDS-PAGE analyses, cTXNPx (50μM) was incubated with different concentrations of compound **1c** (5, 50, 500 μM) for 1 h, followed by incubation with compound **2** (50 μM) for an

additional 1 h. The protein was then submitted to a click reaction with compound **4** (50 μM) as described above and analysed by SDS-PAGE.

For liquid chromatography–mass spectrometry (LC–MS/MS), cTXNPx (1 mg / mL) was incubated with compound **1c** or **3** (1:1) and desalted using Waters MassPrep desalting cartridge. Then, intact proteins were analysed in UPLC system in tandem to LCT premier Acquity (Waters), as described above. Mass operating range was 100–2000 units, protein data was processed using Masslynx 4.1 and deconvoluted using MaxEnt 1 to show the nominal neutral mass of the protein. For thermal shift assay (TSA), 10 μL of cTXNPx (1 mg / mL) in 20 mM HEPES buffer, pH 6.9, 50 mM NaCl and 5 mM DTT were incubated with compound **1c** or **3** at different ratios for 30 min. Then, proteins were transferred into 48-well PCR plates containing 10 μL of SYPRO Orange in water. Fluorescence data were collected using SYPRO Orange wavelengths, 472 / 570 nm on a StepOne qPCR machine (Thermo Fisher Scientific). Raw data was analyzed using NAMI software and Protein Thermal Shift software 1.3 (Thermo-Fisher Scientific) to calculate the derivative curve determined $T_m$ values and to compare $\Delta T_m$ values of different ligands [23, 24].

## Molecular modeling

In the absence of a three-dimensional X-ray crystal structure of cTXNPx from *L. amazonensis* (*La*TXNPx), homology modeling was employed based on the *L. major* crystal structure of the TXNPx homodecamer [25]. The two proteins are highly homologous with a sequence identity of 89%. The template search, selection, model building and quality estimation were conducted altogether with the SWISS-MODEL server [26].

To investigate the binding mode of compound **1a, 1b, 1c** and **3**, respectively within *La*TXNPx, we performed molecular docking studies using the flexible docking approach GOLD [27]. First, the three-dimensional structures of ligands were built and optimized using the Molecular Mechanics Force Field (MMFF)[28] on Spartan'18 software (Wavefunction Inc, 2012) [29]. The protein model was prepared with GOLD with all H-atoms added and charges balanced. The search space was optimized to a radius of 10 A around the peroxidatic cysteine residue Cys 52. The docking process involved high accuracy settings first with GoldScore and rescoring with ChemScore, 30 genetic algorithm runs, and high ligand flexibility. The range of ChemScore fitness was 1.6 to 19.3 (**1a**),1.9 to 14.6 (**1b**), -1.6 to 17.4 (**1c**) and -14.4 to 14.8 (**3**). The best binding pose was identified as the solution with the highest ChemScore fitness value. All figures were created with PyMol (Schrodinger, LCC, 2010, The PyMol Molecular Graphics System, Version 2.1).

## CRISPR-Cas9 cTXNPx gene knockouts

cTXNPx genes deletion was performed as described in Beneke et al [30]. The online primer design tool www.LeishGEdit.net was used to design primers for amplification of the 5′ and 3′ sgRNA templates and for amplification of donor DNA from pT plasmids (S3 Table). *L. mexicana* Cas9 T7 promastigotes [30] (1 x 10$^7$/mL), previously kept in the presence of nourseothricin sulphate (50 μg/mL) and hygromycin B (32 μg/mL), were transfected with 10 μg of each respective donor DNA and sgRNAs using 2 mm gap cuvettes (MBP) with program X-001 of the Amaxa Nucleofector IIb (Lonza Cologne AG, Germany).

Following transfection, parasites were transferred into 5 mL prewarmed medium in 25 cm$^2$ flasks and left to recover overnight at 26˚C. Then limiting dilution, in the presence of appropriate selection drugs, was used to generate clonal tagged cell lines. First, KO parasites for both copies of each gene were selected using different markers, LmxM.15.1040 KO (Δ1040) was selected with blasticidin-S deaminase (10 μg/mL), while LmxM.15.1160 KO (Δ1160) selected

with puromycin dihydrochloride (20 µg/mL). Survival of selected transfectants became apparent 7–10 days after transfection. Deletion of both copies for each gene was confirmed by PCR (S9 Fig). Using the Δ1160 cell line, the remaining gene (LmxM.15.1040) was excluded through a new round of transfection using a neomycin resistance gene as the repair cassette. The final cell line, cTXNPx KO (ΔcTXNPx) was selected with G-418 disulfate (40 µg/mL).

### Diagnostic PCR for knockout verification

Genomic DNA of drug-selected clones was isolated with Wizard Genomic DNA Purification Kit (Promega). To assess the loss of the target gene ORF and the presence of donor DNA in putative KO lines and the parental cell lines, a diagnostic PCR was performed using GoTaq G2 DNA Polymerase (Promega), within 100ng of gDNA as template and forward primer and reverse primer for each ORF or resistance gene.

The PCR conditions were 2 min at 95˚C followed by 35 cycles of 30 s at 95˚C, 45 s at 60˚C, 1 min 72˚C, and a final extension step of 5 min at 72˚C. The presence of expected amplicon was confirmed by 1% agarose gel. Primer sequences are detailed in Supplementary Materials in S4 Table.

### Mutant cell growth

Promastigotes of *L. mexicana* T7 Cas9 (Parental), Δ1040, Δ1160 and ΔcTXNPx, at the second passage, were cultured at 26˚C in Schneider's medium supplemented with 10% HIFBS, and the number of parasites was counted daily for four days using a Neubauer chamber. On day 3 of culture, parasite aliquots were placed onto microscope slides and cells imaged using fluorescence microscope (Nikon-EclipseTi) with a CFI Plan Apo Lambda 100X Oil objective and differential interference contrast (DIC) module.

### Statistical analysis

Single-variance analysis (ANOVA) and *t*-test were used to compare differences between group samples followed by Dunett post-test, for comparisons of all groups to control group. For multiple comparison the post-test used was Bonferroni. All statistical analyses were performed using GraphPad Prism 7 software and differences were considered significant when $p \leq 0.05$, in a number of at least 3 independent experiments.

## Results

### Chemical tools for chalcone target identification

Whilst CH8 (**1b**; Fig 1) showed good activity, drug development was challenged by a difficult synthesis in which flavone formation accompanied aldol condensation. Consequently, a trimethoxylated analogue NAT22 (**1c**; Fig 1), that could be prepared far more efficiently through simple aldol condensation of the commercially available trimethoxyacetophenone with 3-nitrobenzaldehyde, was used as the parent chalcone [31]. Importantly, this showed similar anti-leishmanial activity to chalcone CH8 (**1b**) *in vitro* and *in vivo* against *L. amazonensis* [31].

The highly electrophilic enone functional group within CH8 was predicted to be key to activity [32]. This hypothesis was verified by the lack of a significant antileishmanial effect of dehydrochalcone analogue (**3**) (Fig 2A). To further investigate the mode of action, chalcone **1c** was converted into an activity-based probe **2** (S1 Fig). Briefly, commencing from commercially available 2,6-dimethoxyacetophenone, regioselective tandem Ir-catalysed C-H borylation—oxone oxidation followed by alkylation with 6-bromohex-1-yne and aldol condensation afforded the alkyne tagged chalcone probe **2**. This modification did not alter the intrinsic

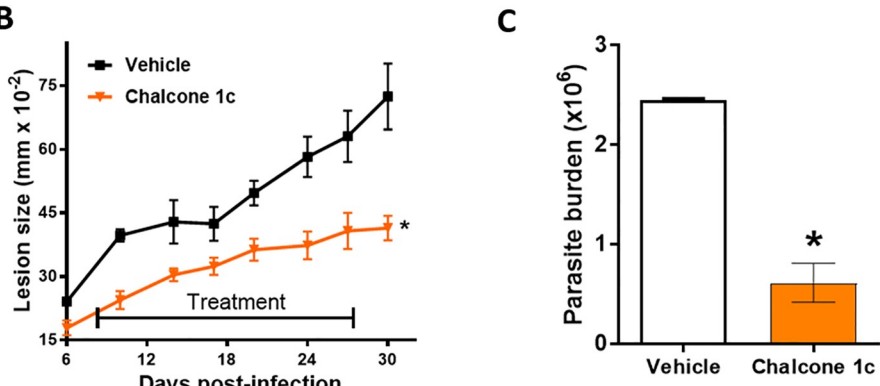

**A**

| Compounds | EC$_{50}$ [μM] Promastigotes | | | EC$_{50}$ [μM] Amastigotes | CC$_{50}$ [μM] Macrophages | Selectivity index |
| --- | --- | --- | --- | --- | --- | --- |
| | *L.amazonensis* | *L.braziliensis* | *L.infantum* | *L. amazonensis* | | EC$_{50}$/CC$_{50}$ |
| 1c | 0.64 ± 0.08 | 1.96 ± 0.12 | 0.76 ± 0.08 | 0.32 ± 0.06 | 7.37 ± 0.06 | 23.03 |
| 2 | 0.42 ± 0.04 | 1.45 ± 0.14 | 0.78 ± 0.07 | NT | NT | NT |
| 3 | 67.13 ± 0.07 | 252.00 ± 0.09 | 26.18 ± 0.07 | 19.57 ± 0.36 | 64.18 ± 0.07 | 3.28 |

**Fig 2. *In vitro* and *in vivo* antileishmanial activity. (A)** *In vitro* activity: Promastigotes of the indicated *Leishmania* species; intracellular amastigotes of *L. amazonensis*, and uninfected macrophages were incubated with different concentrations of **1c**, **2** and **3** for 48 h. Then, cell viability was assessed as described in M&M for determination of EC$_{50}$ and CC$_{50}$ for parasites and macrophages, respectively. *In vivo* efficacy: Mice were infected in the ear with *L. amazonensis*. Seven days after infection, were given chalcone 1c (40 mg/Kg) or vehicle alone (100 μL of propylene glycol) daily for 3 weeks by intragastric gavage. **(B)** Ear thicknesses were measured in the indicated days with a digital caliper. Lesion sizes were the difference between infected and noninfected ears in each time point. **(C)** Ear parasite loads were determined by limiting dilution assay. Means ± SD (n = 5 animals/ group). *P < 0·05 in relation to Vehicle. NT- Not tested.

chalcone activity against the promastigote parasite, with EC$_{50}$ values (*L. amazonensis* 0.42 μM, *L. braziliensis* 1.45 μM and *L. infantum* 0.78 μM) comparable to compound **1c** (Fig 2A–2C).

To interrogate the mechanism(s) by which chalcone **1c** is able to kill *Leishmania* spp., live parasites were incubated with compound **2** (5 μM) and, following conjugation with the tri-functional reagent **4** [33, 34], visualized by confocal microscopy (Fig 3A). The images obtained indicated that the chalcone is widely dispersed in the parasite but is excluded from the nucleus. To further investigate this localisation, promastigotes were lysed, and the soluble protein fraction treated with **2** and then **4**, as above. SDS-PAGE analysis revealed a fluorescent protein band at about 22 kDa (Fig 3B). Competitive dosing with chalcone **1c** led to a reduction in labelling of this band in concentration-dependent manner (Figs 3C and S2). The labelled band was excised from the gel, digested and the tryptic peptides analysed by LC–MS/MS. The comparison of these peptide sequences with the reported genome for *L. amazonensis* [35], revealed 10 candidate proteins (S3 Fig and S1 Table). Comparison of molecular masses for these suggested that cytosolic tryparedoxin peroxidase (cTXNPx, NCBI entry: AAX47428.1) as the predominant chalcone target (Figs 3D and S3). Two-dimensional gel analysis followed by mass spectrometry (S4 Fig and S2 Table) and immunoblotting analysis with anti-cTXNPx which also indicated that cTXNPx was also labelled by chalcone **2** in intracellular amastigotes with no signal being observed in uninfected macrophage samples (S5 Fig).

## Chalcones can form a strong and stable interaction with cTXNPx

To further explore the interaction between cTXNPx and antileishmanial chalcones, *L. amazonensis* cTXNPx (*La*TXNPx) was expressed recombinantly in *E. coli*, purified and subsequently

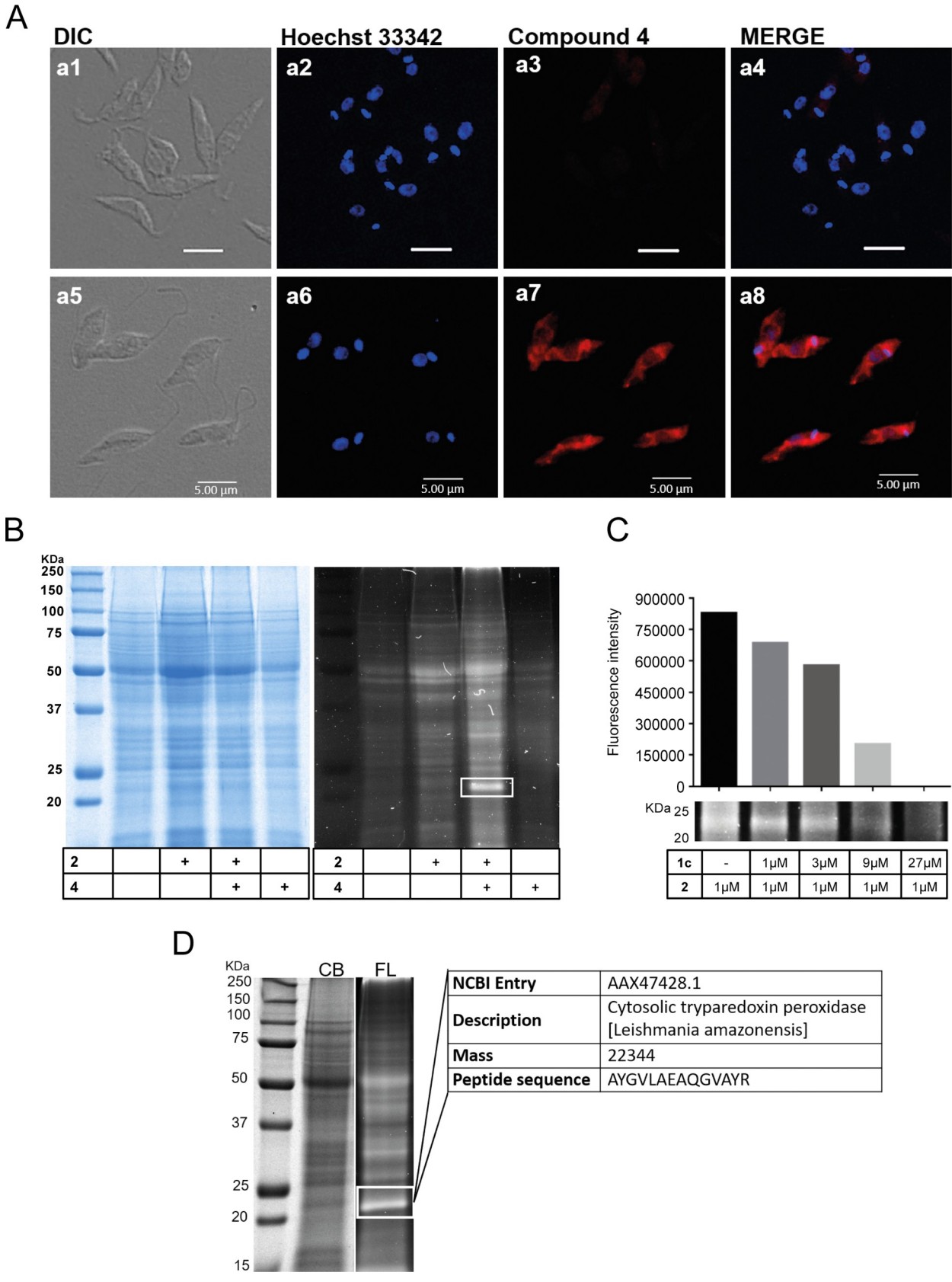

**Fig 3. Intracellular probe binding and identification of cytosolic tryparedoxin peroxidase as protein target.** (**A**) Confocal microscopy of *L. amazonensis* promastigotes treated (a5-a8) or not (a1-a4) with **2** (5 uM) for 5 h prior to labelling with red fluorescent probe **4** (5uM) by CuAAC. (**B**) In gel fluorescence, detection of proteins from *L. amazonensis* promastigote lysate pre-treated with **2** (5uM) for 4 h and linked to molecule **4** by CuAAC. White rectangle- fluorescent band at around 22 kDa. (**C**) In gel representative fluorescent image of competition assay between molecule **1c** and **2**. Promastigotes were incubated with molecule **1c** (1, 3, 9 and 27 μM) for 4 h, washed, and incubated with **2** (1 μM) for additional 4 h. Parasites were lysed, and the lysates were incubated with **4** prior to SDS-PAGE (35 μg / lane).Qualitative decrease in fluorescence signal by addition of increasing amounts of **1c**. (**D**) Identification of cytosolic tryparedoxin peroxidase (cTXNPx) as a best candidate by mass spectrometry analysis of peptides from excised 22kDa band, by ESI-Q-TOF.

incubated with **2**. SDS PAGE analysis following cycloaddition reaction with **4** (50 μM) showed a labelled band at 24 kDa (Fig 4A, L3). As observed for the live parasites, incubation with **1c** inhibited the binding of **2** and the subsequent fluorescent signal of this band in a concentration dependent manner (Fig 4A). Thermal shift assay (TSA) revealed that binding of **1c** led to a reduction in $T_m$ in a dose dependent manner (Fig 4B) indicative of a significant degree of destabilisation of protein structure [36]. Mass spectrometric analysis of cTXNPx treated with chalcone **1c** revealed a new peak at 24368 Da [24025 + 343] (Fig 4C and 4D), corresponding to a stable protein chalcone adduct that suggested that this interaction is based on a strong and stable interaction, very likely a covalent bond. Similar observations could be made using recombinant cTXNPx from *L. major* (LmTXNPx) incubated with compound **1c** (S6 Fig). The degree of labelling is only partial and attempts to drive this to higher levels through longer incubation times were not successful. Evidence that this interaction is necessarily dependent on the enone group, was demonstrated through the use of dehydrochalcone **3**. Using the same sequence of experiments, this compound, even at high concentrations (180 μM—4:1 (ligand: protein)) was not able to affect protein $T_m$ values (Fig 4B) or lead to any evidence for a labelled protein by mass spectrometric analysis (Fig 4E).

## Chalcone 1c leads to ROS accumulation in parasites

cTXNPx belongs to the highly-conserved 2-cysteine peroxiredoxins that have a key role in the protection of the cell from hydroperoxides and peroxynitrites [25, 37]. Collectively, our results suggested that, in *Leishmania* parasites, compound **1c** binds to and inhibits cTXNPx, leading to ROS accumulation and cell death. To confirm this hypothesis, *L. amazonensis* promastigotes were incubated with **1c** or **3**, and ROS production measured by fluorimetry using $H_2DCFDA$ (2,7-dichlorodihydrofluorescein diacetate) (S7A and S7B Fig) [38]. Compound **1c** led to a decrease in the number of parasites in a concentration dependent fashion with parasite killing being accompanied by ROS accumulation (S7A Fig). In contrast, the dihydrochalcone **3**, resulted in a lower parasite killing and, even at higher concentrations, no significant increase in ROS accumulation (S7B Fig). Similarly, in infected, but not in uninfected, macrophages **1c** was able to induce ROS accumulation whilst **3** was unable to affect ROS levels in either infected or uninfected macrophages (S7C and S7D Fig). Importantly, the lack of a response on uninfected macrophages to treatment with **1c** or **3** indicated that the increased ROS levels were attributable to intracellular amastigotes (S7D Fig).

## Chalcones bind in the cTXNPx active site

To better understand the molecular mechanism of the interaction between the anti-leishmanial chalcones and cTXNPx, we examined the interaction by molecular modelling. Using the amino-acid sequence of *La*TXNPx (UniProtKB database, access code Q4VKK8) [39, 40] and the crystal structure of *Lm*TXNPx [41], which have 89% sequence identity, a homology model was built using the SWISS-MODEL server.

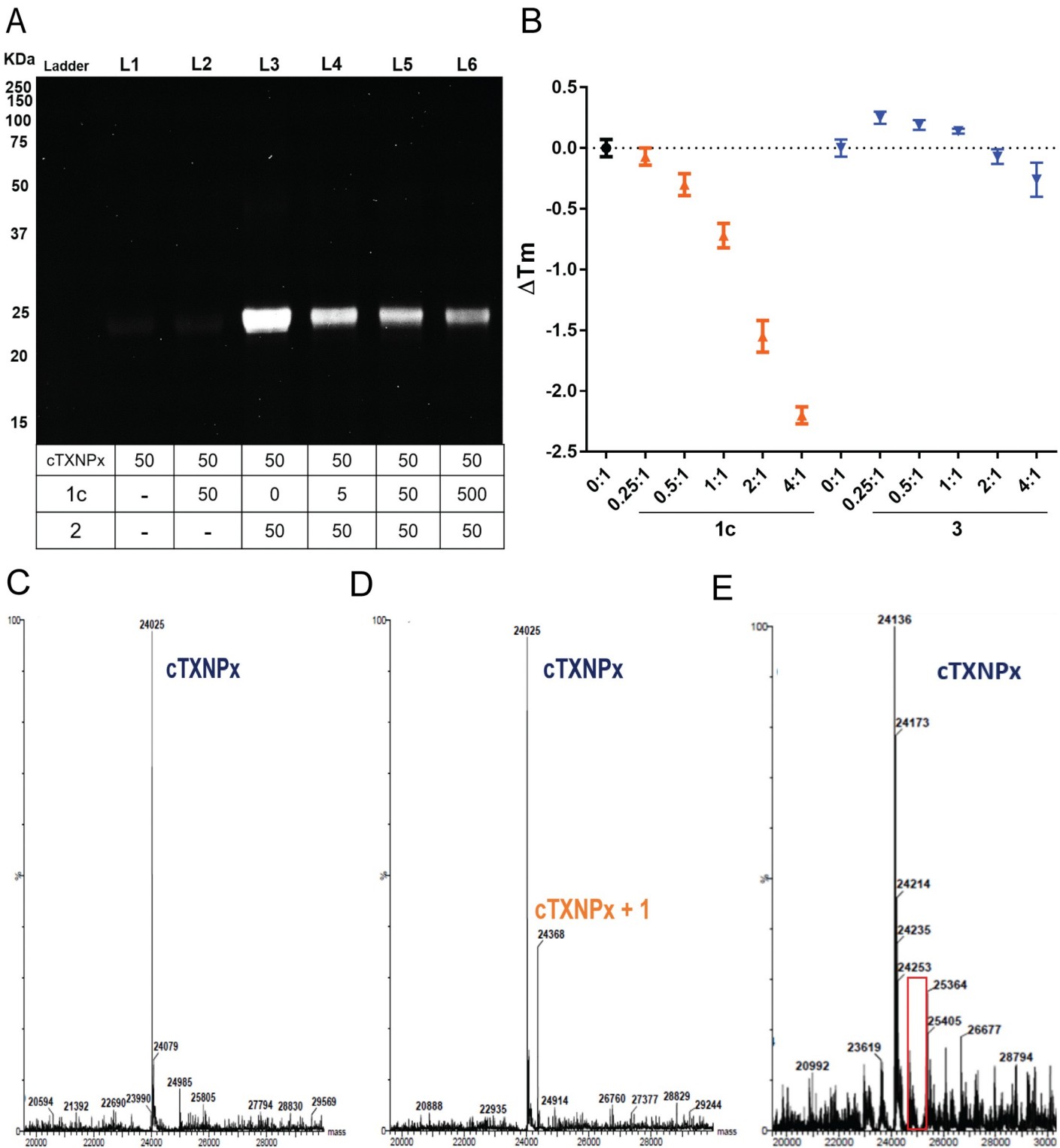

**Fig 4. Chalcone impairs the cTXNPx antioxidant effect in the parasite by a stable interaction mediated by enone group.** (**A**) Competition binding assay between molecules **1c** and **2** employing recombinant cTXNPx. Recombinant *L. amazonensis* cTXNPx was incubated with the indicated concentrations of 1c for 60 min, then washed, incubated with **2** (50 μM), and linked to **4** by CuAAC. Representative fluorescence gel images signals of **4**. (**B**) protein melting curve from thermal shift assay of cTXNPx protein incubated with the indicated concentrations of **1c** (brown) or **3** (blue). (**C-E**) MS analysis of interaction between chalcone and cTXNPx. Purified proteins (0.6 mg/ mL) were incubated or not with compound **1c** or **3** (1:1) for 30 min and their molecular weight analysed by MS. (**C**) cTXNPx of *L. amazonensis* [M-Met requires 24025 Da]. (**D**) cTXNPx of *L. amazonensis* plus 1 [M-Met+343 requires 24368 Da]. (**E**) cTXNPx of *L. major* plus 3 [M-2H-Met requires 24136 Da]. Means ± SD (n = 3 independent experiments). * $p < 0.05$ in relation to untreated cells.

In common with most peroxiredoxins, this *La*TXNPx model was obtained in the fully folded (FF) conformation as a homodecamer (Fig 5A) that can also be described as a pentamer of dimers (Fig 5B). The active site is formed by the N-proximal peroxidatic cysteine (Cys52) [42] from one subunit and a C-proximal cysteine from the two-fold symmetry related subunit (Cys173). These residues form a disulfide bridge in the oxidized state [43]. It is important to note that all crystal structures, as well as the homology model, are in the reduced form where the two cysteine residues are approximately 10 Å apart. Docking studies on the reduced homo-dimer, focussing on the critical Cys52, were performed to investigate possible binding modes of the chalcones. The dimer interface exhibits a narrow groove at the end of the helix harbour-ing the active Cys52 where all chalcones can bind as shown for **1a** in Fig 5C. The chalcones **1b** and **1c** (Fig 5D and 5E) showed a similar binding pose with the phenyl and nitro-phenyl group on the same side of the tight groove where the enone moiety is located, approximately 7.7 Å from the sulphur atom of Cys52. Thus, only a relatively small rearrangement is required to allow the formation of the covalent adduct. This may explain why only partial covalent protein modification is observed. While compound **3** can also occupy the same pocket as shown in Fig 5F, its binding poses show significantly lower docking scores compared to the chalcones and the conformation is clearly more strained.

## Chalcone antipromastigote activity is dependent on cTXNPx

In order to confirm the cTXNPx enzyme as a chalcone target as well as to examine its impor-tance for the parasite biology, we generated parasites lacking cTXNPx genes. Due to the avail-ability of a well described and utilised CRISPR-Cas9 system giving a straightforward way to delete genes in *L. mexicana*, we choose this platform to explore these questions [30, 44].

cTXNPx is encoded by two genes (LmxM.15.1040 and LmxM.15.1160), located at chromo-some 15, sharing 98% and 97% of similarity at the gene and protein level respectively (S8 Fig). To analyse the potential role of each in parasite fitness and chalcone sensitivity, using sgRNAs specific for the flanking regions (UTRs) we generated parasites with each gene knockout out separately. In this experiment LmxM.15.1040 alleles were replaced by a repair cassette encod-ing the blasticidin resistance maker (Δ1040) and, separately, LmxM.15.1160 by the puromycin resistance marker (Δ1160). Knockouts for each were selected following cloning and PCR anal-yses (S9 Fig). Subsequently, using the same approach, commencing from the Δ1160 cell-line, the remaining gene was replaced by a sequence encoding neomycin resistance to generate a line in which both LmxM.15.1040 and LmxM.15.1160 were ablated in this diploid parasite (ΔcTXNPx) (Fig 6A and 6B). The correct integration of the puromycin, blasticidin and neo-mycin-resistance markers in the cTXNPx locus, and the complete loss of cTXNPx, were con-firmed by PCR analyses as shown in Figs 6B and S9. No growth defect or morphological alterations, in comparison to the parental line, were observed when only one cTXNPx gene (Δ1040 or Δ1160) was deleted (Fig 6C and 6D). However, the absence of both genes (ΔcTXNPx) caused a severe alteration in parasite growth in comparison to the other lines (Fig 6C) and significant phenotypic alterations when observed by optical microscopy (rounded cell body and short flagellum; Fig 6D).

To investigate if the specific cTXNPx enzyme is the chalcone target *in cellulo*, promastigotes of parental and mutant cell lines (Δ1040, Δ1160 and ΔcTXNPx) were incubated with different concentrations of compound **1c** for 48 h when the parasite viability was assessed using alamar-Blue (Fig 6E). As expected, compound **1c** showed a good activity against the parental line (EC$_{50}$ 1.04 μM), similar to the values found against other parasite species (Fig 2B). The lack of one of Δ1040 or Δ1160 cell lines reduced the EC$_{50}$ 2 and 1.6-fold respectively (EC$_{50}$ 1.99 μM and 1.67 μM; Fig 6E). When both genes were deleted in ΔcTXNPx cell line, the parasite was

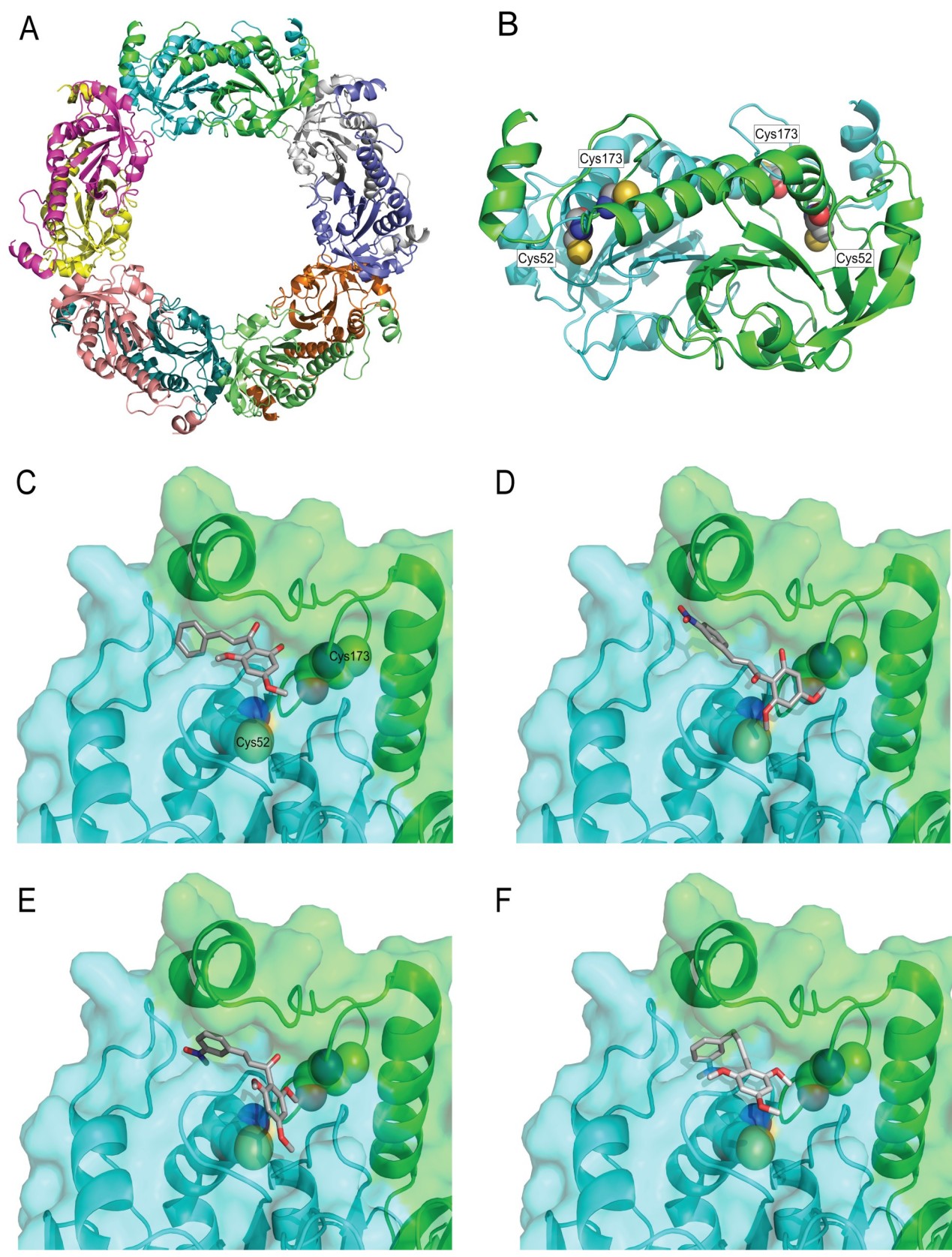

**Fig 5. *La*TXNPx 3D model and 3D/2D binding mode of 1c and 3.** (**A**) Ribbon diagram of the cTXNPx homology model. Each chain of the homodecamer is depicted in a single colour. The decamer can be described of pentamers of functional homodimers. (**B**) Ribbon diagram of one dimer with each chain depicted in green and cyan, respectively. The key cysteine residues 52 and 173 are depicted in CPK representation with C-atoms on grey, O-atoms in red, N-atoms in blue and S-atoms in yellow. Note, that the cysteines are present in their reduced forms. H-atoms are omitted for clarity. (**C**) Close-up of the proposed binding site with the highest scoring binding pose of compound **1a** depicted in a stick-representation with C-atoms in grey and O-atoms in red. (**D**) highest scoring binding pose of **1b** (**E**) **1c** (**F**) **3**.

3.8-fold more resistant to the chalcone, with $EC_{50}$ = 3.87 μM (Fig 6E). Together, these results indicate that cTXNPx is important for the parasite biology and also likely to represent a major target of antileishmanial chalcones.

## Discussion

Conventional chemotherapy employed in leishmaniasis treatment relies on only a few drugs, all of which have limitations including cost, toxic side effects, low or variable efficacy that

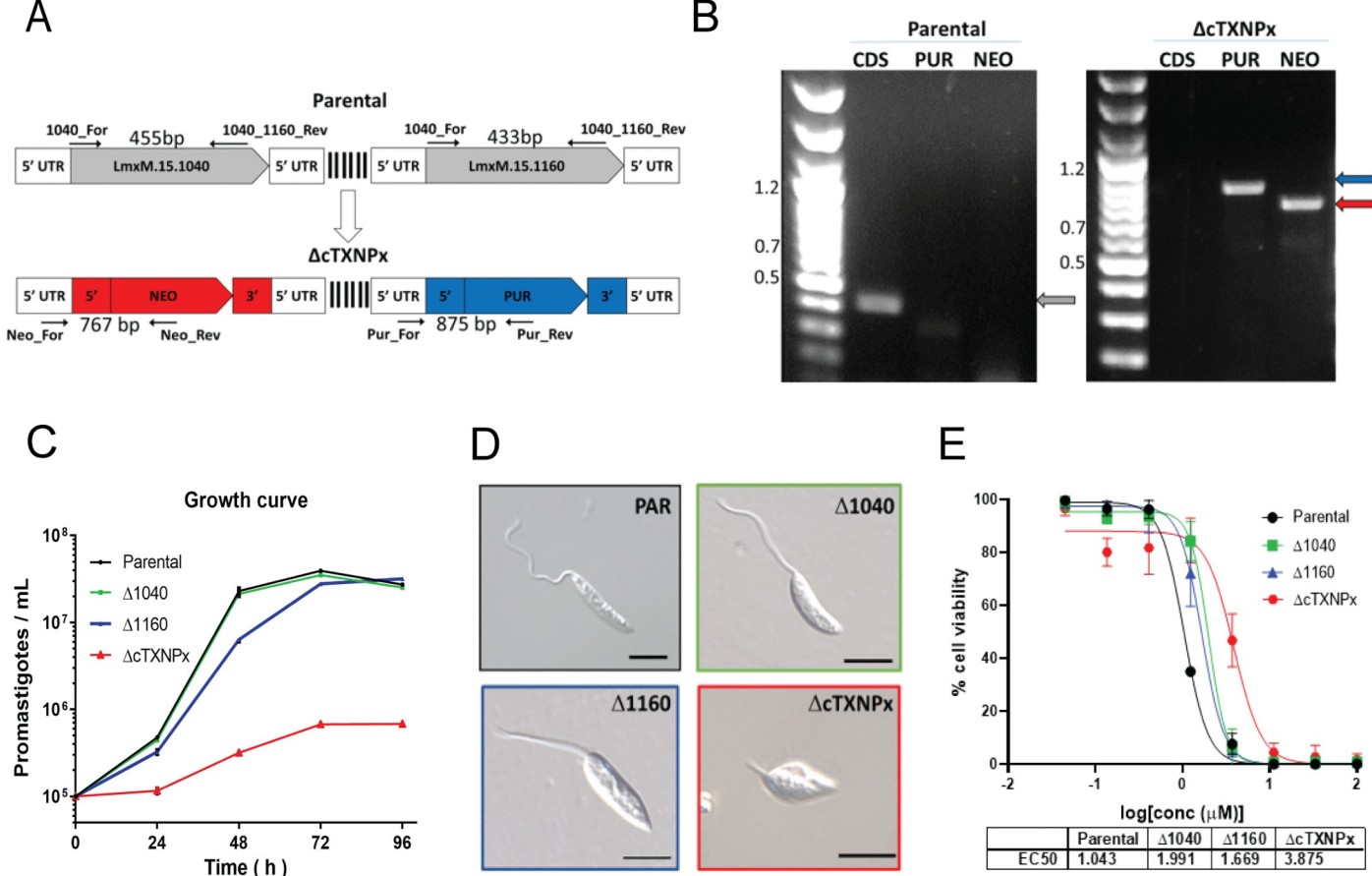

**Fig 6. Chalcone antiparasitic activity is partially dependent on cTXNPx.** (**A**) Diagrams showing the PCR strategy to confirm the replacement of LmxM.15.1040 with the neomycin (NEO) cassette in the LmxM.15.1160 double KO (puromycin [PUR] cassette; S9 Fig) to create ΔcTXNPx. (**B**) PCR analysis of the ΔcTXNPx cell line, PCR products visualized on agarose gels. Coding sequence for LmxM.15.1040 (CDS, grey arrow), puromycin (PUR, blue arrow) and neomycin (NEO, red arrow) in Parental and ΔcTXNPx cells. (**C**) Promastigotes were seeded at $1x10^5$ parasites/mL and the number of parasites in culture were counted every day for 4 days using a Neubauer hemocytometer in triplicate from two independent experiments. Cell lines: Parental (black line), LmxM.15.1040 double KO (green line), LmxM.15.1160 double KO (blue line), and cTXNPx KO (red line) (**D**) Representative Differential interference contrast micrographs showing all cTXNPx mutant cell lines analysed in **C** and **E**. Scale bar, 5 μm. (**E**) cTXNPx mutant promastigotes ($2x10^6$/mL) were incubated with different concentrations of **1c** for 48h. Cell viability was then assessed by the addition of resazurin solution for 4 h and $EC_{50}$ values calculated by non-linear regression. Means ± SD (n = 3 independent experiments).

depends on parasite species, difficult modes of administration and growing parasite resistance [5]. This current scenario clearly points to the urgent demand for new treatments and innovative approaches to control this neglected tropical disease (NTD) which affects more than one million people worldwide per year [2]. Strategies for the discovery of new drug leads can broadly be partitioned into target-based and phenotypic approaches [45]. For NTDs such as leishmaniasis, the latter has become the favoured approach with high-content high-throughput screening studies being described enabling the broad diversity of structures found in small molecule and natural product collections to be explored without preconceptions [46, 47]. However, determination of the molecular target and mode of action is then required to facilitate further drug development [48–50].

Previously, we described the discovery and development of the synthetic nitro-chalcone **1b** (CH8), to effectively treat both CL and VL in an animal model [13–16]. However, as with many such studies, despite excellent results against the parasite, the molecular target and mechanism of action important for the development of new chalcone leads remained unknown [9]. To address this, we have employed a chalcone derived activity-based chemical probe (ABPP) based approach, to identify cytosolic tryparedoxin peroxidase (cTXNPx) as a target of this compound. Whilst ABPP has been employed quite widely in cell biology, application to *Leishmania* spp. is limited [51–57]. Similarly, whilst many bioactive natural products have been adapted for use as ABPP agents to identify and validate new drug targets [58], despite being intrinsically electrophilic and therefore able to label a protein target, the use of chalcones in this context is surprisingly limited [32, 59].

Belonging to the ubiquitous 2-Cys family of antioxidant enzymes that are highly conserved among trypanosomatids with no direct mammalian orthologue [60], the tryparedoxin peroxidases have key roles in parasite survival and virulence [37, 61]. Their primary function is to reduce reactive oxygen species (ROS), most notably generated in the oxidative burst that follows parasite internalisation by macrophages. The ultimate electron donor in this cycle is tryparedoxin which distinguishes *Leishmania*, and other kinetoplastids, from most other eukaryotes, including the mammalian host, where thioredoxin is employed. As such, TXNPx provides an essential mechanism for trypanosomatids to survive under these conditions of biotic stress [62, 63]. Moreover, TXNPx has been shown to be a major contributor to the growth of resistance to many of the existing drugs [64, 65], with over-expression in *L. donovani* and *L. amazonensis* providing protection against oxidative stress from macrophages as indicated by increased parasite infectivity [37, 66].

Reflecting this unique and species-specific role, TXNPx may be considered a potential drug target [67]. Significantly this hypothesis is supported by our results in which genes encoding cTXNPx were deleted, leading to parasites with a compromised phenotype. Moreover, the lower efficacy of the chalcone in these mutants clearly demonstrated the involvement of cTXNPx in the antileishmanial activity of these nitro chalcones. However, the fact that significant activity remains suggests that the chalcones have multiple modes of action that are working in concert. Significantly, in this context it is pertinent to note that *Leishmania* spp. possess two distinct tryparedoxin peroxidases, a mitochondrial and cytosolic form, sharing approximately 50% similarity at protein level and with a conserved tertiary structure (S8 Fig) [61]. Interaction of the chalcones with the mitochondrial tryparedoxin peroxidase would be consistent with the earlier reports of mitochondrial alterations on treatment with DMC (**1a**) and cannot be ruled out at this stage [11]. The fact that no labelling of this second tryparedoxin peroxidase was detected in our experiments may simply reflect its lower cellular abundance and the low levels of protein alkylation observed (Fig 4D) in our experiments.

The pull-down experiment reported here is consistent with a model in which the nitro chalcone and cTXNPx interplay occurs through a covalent bond, potentially the Michael addition

of a cysteine thiol with the electrophilic enone group [32, 68]. Similar mechanisms of inhibition of thioredoxin reductases by chalcones [69] and other Michael acceptors, such as curcumins, cinnamaldehydes, and fungal metabolites [70–72] have been described. This proposal is supported by our studies with compound **3** where the lack of the α,β-unsaturation resulted in loss of chalcone interaction with cTXNPx protein *in vitro* and *in situ*, and consequently low antiprotozoal efficacy. However, despite the ability to isolate cTXNPx by pull-down, the labelling of the protein is not complete, and we were not able to identify the specific amino acid residues involved in this process. We suggest that this partial labelling is indicative of a two-stage process involving initial binding of the chalcone into active site followed by a slower non-specific alkylation event, potentially following disruption of the critical active dimer.

Support for this pathway comes from our molecular modelling and docking studies which reveal how **1c** can access the enzyme active site, fitting between Cys-52 and Cys-173, and block disulfide bond formation. This orientation of **1c** places the electrophilic carbon of the enone in close proximity (8.0 Å) to the reactive Cys-52 residue. Monitoring this interaction by thermal shift assay (TSA) revealed that binding of **1c** leads to a lower melting temperature, suggesting that binding was accompanied by a partial unfolding of the protein [36, 73]. Again, this proposal is supported by the observation that the 'inactive' dihydro analogue **3** exhibited less favourable binding in the docking studies and did not affect protein stability. Consequently, whilst the α,β-unsaturated ketone is essential for ABPP activity, this suggests that the double bond has a conformational role in inhibition of enzyme activity. Importantly, this implies that alternative biosteres can be designed to avoid the potential problems associated with the generic electrophilicity and potential polypharmacology of the chalcone template.

Based on these findings, we propose a mechanism of action for the nitrochalcones. Chalcone **1c** is taken up by the macrophage and is able to access the amastigotes. Here, **1c** interacts with cTXNPx, enzyme recycling is impaired and ROS levels, especially $H_2O_2$ and $ONOO^-$, rise within the amastigote leading to apoptosis-like cell death due to oxidative damage to lipids, proteins and nucleic acids [74–76]. In support of this hypothesis, many indications of apoptosis-like cell death, such as mitochondrial damage [11], DNA fragmentation and other morphological changes have been observed in *Leishmania* spp. on chalcone treatment.

Collectively, these results demonstrate that the nitrochalcone **1c** interacts with cTXNPx leading to protein deformation and inactivation and ultimately parasite death due to cellular damage by non-scavenged ROS. Whilst many of the existing antileishmanial drugs, like pentavalent antimonials, also lead to elevated levels of ROS these effects are non-specific and contribute to the off-target toxicity associated with their use. Moreover, resistance to these chemotherapies is rising and this can be associated with increased tolerance to ROS. It is therefore significant that nitrochalcone **1c** targets a parasite specific protein that contributes to this defence mechanism. As such **1c** not only represents a valuable new drug lead, but also potentially enables the design of combination therapies with the added benefits of reducing therapeutic dose and related toxic effects, and retarding resistance mechanisms [77, 78]. Studies towards this goal are in progress and will reported in due course.

## Conclusion

In conclusion, this work validates TXNPx as a much-needed drug target for the treatment of leishmaniasis. Moreover, the chalcone **1c** represents a well-defined structural starting point in the search for new antileishmanials. As such, this work opens a new perspective in the search for treatments for leishmaniasis, an important but neglected tropical disease which impacts millions of people worldwide and has a high economic and social impact.

## Supporting information

**S1 Fig. Synthetic routes for preparation of compounds 2–3.**
(TIF)

**S2 Fig. Uncropped SDS-PAGEs corresponding to Fig 3C. (ai)** SDS-PAGE of all tested conditions stained with comassie blue. **(aii)** Fluorescence image of all tested conditions.
(TIF)

**S3 Fig. Uncropped SDS-PAGEs corresponding to Fig 3D. (a)** SDS-PAGE stained with Coomassie blue. **(b)** In-gel fluorescence image. Yellow rectangles- Protein bands excised for mass spectrometry analysis. **(c)** Proteins candidates identified from the band.
(TIF)

**S4 Fig. Confirmation of cTXNPx as a chalcone target by two-dimensional gel analysis. (a)** Map of the dots excised from the 2D gel stained with comassie blue and submitted to mass spectrometry analysis. **(b)** In-gel fluorescence detection image. **(c)** Proteins candidates identified for each 2D dot.
(TIF)

**S5 Fig. Immunoblotting analysis of chalcone labelling on cTXNPx in different parasite stages.** Proteins from promastigotes (Pro), isolated intracellular amastigotes (Ama), and macrophage (Mac) lysates that had been pre-incubated with **2** (5 μM) and linked to **4** (5 μM) prior to fluorescence revelation. Alternatively, the run lysates were revealed with mouse anti-cTXNPx antibody followed by anti-Mouse IgG-HRP (anti- cTXNPx). a. In-gel fluorescence detection image. b. Immunoblotting analysis with anti-cTXNPx. Proteins from promastigotes (P1), amastigotes (A1) and macrophages (M1) without **2**. Proteins from promastigotes (P2), amastigotes (A2) and macrophages (M2) pre-incubated with **2.**
(TIF)

**S6 Fig. Mass spectrometry analysis of interaction between chalcone and Lm-cTXNPx.** Purified proteins (0.6 mg/ mL) were incubated or not with compound **1c** (1:1) for 30 min and their molecular weight analysed by Mass Spectrometry. **(a)** cTXNPx of *L. major* molecular weight [M-Met requires 24138 Da]. **(b)** cTXNPx of L. major with **1** [M-Met+343 requires 24481 Da].
(TIF)

**S7 Fig. Inhibition of cTXNPx by chalcone impairs the enzyme antioxidant effect in the parasite.** *L. amazonensis* promastigotes ($1 \times 10^6$/mL) were incubated with compounds **1c** or **3** at the indicated concentrations for 48 h, during which the numbers of cells were recorded. Then, $4 \times 10^5$ cells were transferred to black 96 well plates for fluorimetric assessment of ROS production with $H_2DCFDA$ (20 μM) for 30 min. **(a-b)** Parasite killing (dotted lines) and ROS production (continuous lines) induced by 1c (a) 3 **(b).** BMDM ($1 \times 10^5$/well) infected or not with L. amazonensis promastigotes (10:1) for 72 h on 96 well-plate, then cells were incubated with 1c or 3 (0.07, 0.7 and 7.0 μM) for 1 h at 37°C and ROS production measured as described for promastigotes. (c) ROS production on infected macrophages. (d) ROS production on uninfected macrophages. Means ± SD (n = 3). $^*$ $p < 0.05$ in relation to untreated cells.
(TIF)

**S8 Fig. TXNPx sequence alignment.** Gene and protein sequences of mitochondrial (LmxM.23.0040) and cytosolic (LmxM.15.1040 and LmxM.15.1160) TXNPx were obtained from https://tritrypdb.org website and aligned using Jalview software (2.11.1.3). (**a**) Gene sequence alignment (**b**) Protein sequence alignment.
(TIF)

**S9 Fig. Screening of KO clones for each gene. (a)** Diagrams showing PCR strategy for assessing presence of gene copies for LmxM.15.1040 and LmxM.15.1160 and their replacement with the blasticidin [BLA] or puromycin [PUR] cassette. **(b)** PCR of clonal lines following transfection with sgRNA targeting LmxM.15.1040 gene with BLA as the repair cassette. Yellow star highlights clonal double KO of LmxM.15.1040 (left; green arrow BLA and grey arrow LmxM.15.1040 CDS) and the retention of LmxM.15.1160 (right; yellow arrow LmxM.15.1160 CDS). **(c)** PCR of clonal lines following transfection with sgRNA targeting LmxM.15.1160 with PUR as the repair cassette. White star highlights clonal double KO of LmxM.15.1160 (right; blue arrow PUR and yellow arrow LmxM.15.1160 CDS) and the retention of LmxM.15.1040 (grey arrow LmxM.15.1040 CDS). P—positive control. Pop—population. (TIF)

**S1 Table. Proteins candidates identified from the band on 1D-gel in S3 Fig.** (XLSX)

**S2 Table. Proteins candidates identified for each 2D dot from S4 Fig.** (XLSX)

**S3 Table. Primers used to knockout cTXNPx in *L. mexicana* using pT plasmids.** (XLSX)

**S4 Table. Primers used for the validation of cTXNPx knockouts.** (XLSX)

**S1 Text. Synthetic procedures.** (DOCX)

## Acknowledgments

We thank Professor Angela Kaysel Cruz (USP-RP) for kindly providing antibody against cTXNPx.

## Author Contributions

**Conceptualization:** Douglas O. Escrivani, Marjolly B. Caruso, Gabriela A. Burle-Caldas, Maria Paula G. Borsodi, Paul W. Denny, Bartira Rossi-Bergmann, Patrick G. Steel.

**Formal analysis:** Douglas O. Escrivani, Rebecca L. Charlton, Marjolly B. Caruso, Gabriela A. Burle-Caldas, Maria Paula G. Borsodi, Natalia Arruda-Costa, Marcos V. Palmeira-Mello, Jéssica B. de Jesus, Ehmke Pohl, Paul W. Denny, Bartira Rossi-Bergmann, Patrick G. Steel.

**Funding acquisition:** Bartira Rossi-Bergmann, Patrick G. Steel.

**Investigation:** Douglas O. Escrivani, Rebecca L. Charlton, Marjolly B. Caruso, Gabriela A. Burle-Caldas, Maria Paula G. Borsodi, Natalia Arruda-Costa, Stefanie Freitag-Pohl, Ehmke Pohl.

**Methodology:** Marcos V. Palmeira-Mello, Jéssica B. de Jesus, Stefanie Freitag-Pohl, Ehmke Pohl, Patrick G. Steel.

**Project administration:** Patrick G. Steel.

**Supervision:** Russolina B. Zingali, Alessandra M. T. Souza, Bárbara Abrahim-Vieira, Ehmke Pohl, Paul W. Denny, Bartira Rossi-Bergmann, Patrick G. Steel.

**Writing – original draft:** Douglas O. Escrivani, Rebecca L. Charlton, Marjolly B. Caruso, Gabriela A. Burle-Caldas, Maria Paula G. Borsodi, Paul W. Denny, Bartira Rossi-Bergmann, Patrick G. Steel.

**Writing – review & editing:** Paul W. Denny, Bartira Rossi-Bergmann, Patrick G. Steel.

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
