## [Decision Letter · Decision Letter 0]

10 Aug 2021

Dear Prof Steel,

Thank you very much for submitting your manuscript "Chalcones Identify cTXNPx as a Potential Antileishmanial Drug Target" for consideration at PLOS Neglected Tropical Diseases. As with all papers reviewed by the journal, your manuscript was reviewed by members of the editorial board and by several independent reviewers. In light of the reviews (below this email), we would like to invite the resubmission of a significantly-revised version that takes into account the reviewers' comments. 

We cannot make any decision about publication until we have seen the revised manuscript and your response to the reviewers' comments. Your revised manuscript is also likely to be sent to reviewers for further evaluation.

Sincerely,

Tiago Donatelli Serafim

Associate Editor

Laura-Isobel McCall

Deputy Editor

Reviewer's Responses to Questions

**Key Review Criteria Required for Acceptance?**

**Methods**

-Are the objectives of the study clearly articulated with a clear testable hypothesis stated?

-Is the study design appropriate to address the stated objectives?

-Is the population clearly described and appropriate for the hypothesis being tested?

-Is the sample size sufficient to ensure adequate power to address the hypothesis being tested?

-Were correct statistical analysis used to support conclusions?

-Are there concerns about ethical or regulatory requirements being met?

Reviewer #1: The methods described in this study were appropriated and adequately described in the manuscript. The samples size (particularly for in vivo essays) was adequate and answered the hypothesis of the authors. Finally, there is no concern about ethical requirement.

Some minor points:

- In the Anti-promastigote activity, please include the growth phase of parasites used in the assays (page 6, lines 111-120).

- Please, indicate the number of mice used for in vivo experiment, using chalcone 1c (page 6, lines 148-160).

- Table S2, Neo-forward primer is missing in the Table.

Reviewer #2: The authors clearly designed a methodology that attended the main questions regarding the purpose of the work: the elucidation of a mechanism of action of chalcones that results in their antileishmanial effect. The in vivo analyses were made under ethic procedures validated by the ethic committee, as described in the methodology. At least 3 independent experiments were performed and the statistical analysis is proper, allowing to address the objectives. The use of competition assay, ABPP, and Leishmania cTXNPx knockouts supported by docking analysis were very important to confirm the hypothesis that TXNPx is a potential target for chalcones in the treatment of leishmaniasis.

Reviewer #3: Methods are sound and well explained. There is only one points I would like the authors to address: For mice infections and treatment, the authors must add to the method section who many animals were used per group/experiment, how many times the experiments were repeated and which statistical methods were used to decide the number of animals used to achieve statistical significance.

**Results**

-Does the analysis presented match the analysis plan?

-Are the results clearly and completely presented?

-Are the figures (Tables, Images) of sufficient quality for clarity?

Reviewer #1: Results are well presented throughout the manuscript. The figures, in general, are clear and have good and enough quality for the reader.

Some minor points:

- Figure 2A is a Table, so please replace it for a Table in the final version of the manuscript (page 27).

- Figure 2C, include the standard deviation of the limiting dilution assay (vehicle group) (page 27).

- Figures S3-C and S4-C should also be replaced by supplementary Tables.

Reviewer #2: The results are very clear and describe efficiently the analysis plan. The figures are sufficiently clear and well delineated.

Reviewer #3: Probe displacement assay - Are 2 and 4 permeable to life cells? If so, I would suggest to show the dose dependent probe displacement assay in live cells to prove target engagement of cTXNPx by 1c, and for future campaigns for the target. The assay will also allow for further calculations of inhibition as Ki and Kd of the compounds. The assay could be anyway developed for the purified recombinant protein to gain information on the kinetics of association/dissociation of the compound with the target. Although the gel-based assay is informative, it is hardly quantitative and is limited to the target binding.

Essentiality - The authors had shown that compound 1c binds to cTXNPx and defend it to be the molecular target responsible for its toxicity to Leishmania. These results go into conflict with the data showing that the genes composing the complex are dispensable in promastigotes. Also, the authors did not provide any evidence of the requirement of the genes to amastigotes. I highly recommend for the mutants to be also investigated as amastigotes in macrophages and/or mice infections.

**Conclusions**

-Are the conclusions supported by the data presented?

-Are the limitations of analysis clearly described?

-Do the authors discuss how these data can be helpful to advance our understanding of the topic under study?

-Is public health relevance addressed?

Reviewer #1: The conclusions of the manuscript are clear and well supported by the results.

Reviewer #2: The authors conclude that the work validates TXNPx as a much-needed drug target for the treatment of leishmaniasis. Really, the current treatment possibilities have limitations, such as high prices, toxicity, and parasitic resistance. An important observation that could be better used to reinforce the importance of this work is that tryparedoxin peroxidases, such as TXNPx, are highly conserved among trypanosomatids, without direct mammalian orthologue. This suggests that chalcones could be a leishmanicidal drug with a low possibility of side effects in mammals.

As commented in the text, leishmaniasis is a disease that impacts millions of people worldwide and has a high economic and social impact. The public health relevance is adressed, considering the problems related to existing drugs and the fact that leishmaniasis are neglected diseases that can be disfiguring or fatal, depending on the clinical form presented.

The authors seem quite sure of the limitations of their analyses, and do not jump to conclusions about their findings.

Reviewer #3: Mode of action proposed - The authors discuss the role of cTXNPx in ROS resistance and suggest its inhibition as the mode of action for 1c, but do not provide any evidence for increase of ROS mediated cell death after treatment. These are well stablished assays that should be performed by the authors for better understanding and support of the mode of action here proposed.

**Editorial and Data Presentation Modifications?**

Reviewer #1: (No Response)

Reviewer #2: Minor issues:

1. Line #38: “Instituo” – Instituto;

2. Line #86: The period is missing after “antileishmanials”;

3. Line #87: Remove the comma;

4. Lines #92-93: “...which showed strong and selective anti-parasite activity in vitro and in vivo against both CL and VL”. Replace CL and VL with the Leishmania species;

5. Line #165: cite the permeabilizing agent used;

6. Line #309: “Single-variance analysis (ANOVA) and t-test was used...” – Single-variance analysis (ANOVA) and t-test were used...;

7. Line #318: “Whilst CH8 (1b; Fig 1) showed good activity”. What good activity? Against what? Same for the line #327;

8. Line #934: Change Leishmania to Leishmania in italic.

- Revise the English with a mother tongue specialist.

- In general, the text has to be revised considering spacing of characters, punctuation, and some other spelling details. For example: 

. Line #111: “5 x 105/mL” instead of “5 x 105 / mL”; 

. Line #114: “72 h” instead of “72h”; 

. Line #122: “26 °C” instead of “26° C”; 

. Line #155: “of growth [16].” instead of “of growth.[16]”;

. Line #181: “1 h” instead of “1 hr”;

. Line #190: “1 h at room temperature” instead of “1 hr at RT”;

. Line #574: “leishmaniasis” instead of “leishmanaisis” ...

Reviewer #3: Line 86 - Missing the period at the end of the phrase. 

Line 148 - Please indicate the number of animals used for mouse infection and treatment/parasite burden. Please provide the statistical method used to determine the number of animals necessary to achieve a reliable assessment. Also, in Fig. 2C provide a figure with results for each mouse and the exact p value obtained for the assay. Also, does the result represent independent assays? Does n=5 mean 5 animals/assay?

Line 284 - Please correct prefix (LmxM) of gene ID for LmxM.15.1040 and LmxM.15.1160 (also in other parts of the text, as line 432).

Line 335 and Figure 2A - The authors mix the use of IC50 and EC50 for cellular based assays in the text and Figure 2A, as well as in other parts of the text in experiments that have a similar read-out. Please consolidate the use of EC50 to report results based on cell death response to compound treatment.

Line 346 - Please insert space after Fig 3C.

Line 451 - The statement ‘To investigate if the specific cTXNPx enzyme is the chalcone target in cellulo’ must be rethought and rephrased since the method used by the authors only assess the cell viability after treatment, not the target engagement.

**Summary and General Comments**

Reviewer #1: In this manuscript, the authors described the activity in vitro and in vivo of a nitroanalogue (compound NAT22, 1c) against Leishmania species. This compound is a derivative of a natural chalcone with previously reported activity against Leishmania in vitro and in vivo. The authors also described that the cytosolic tryparedoxin peroxidase (cTXNPx) is the main target of this compound, leading to ROS accumulation and parasite death. These findings were confirmed by deletion of cTXNPx genes by CRISPR-Cas9 technology. Finally, molecular docking studies confirmed that the compound 1c is able to bind the putative active site of cTXNPx.

In general, the manuscript is clear and well written; however some points should be considered for publication.

Main points:

- Concerning to the diagnostic PCR for mutant cell lines of cTXNPx genes, did the authors only investigate for LmXm.15.1160 coding sequence, as indicated in the Figure 6A (right panel)? What about the other copy (LmXm.15.1040 gene)? In Table S2, it is indicated that there are specific primers for each one of these genes.

- Please, include in the Figure 6A, the PCR product expected for LmXm.15.1040 coding sequence.

- It is also missing the diagnostic PCR for ORF of cTXNPx genes in each one of mutant cell lines (Δ1040 and Δ1160), using parental line as control. Please include it in the manuscript.

- Were mutant cell lines (Δ1040 and Δ1160) generated by loss of heterozogosity? These mutant cell lines only have one drug resistance marker. Please specify this point in the manuscript.

- In the agarose gel for ΔcTXNPx line, it is missing the PCR for blasticidin S deaminase gene.

- Please, include standard deviation of EC50 values of the mutant cell lines (page 31, Fig. 6D).

- I would also suggest to present the graphical abstract as a Figure in the manuscript.

- Line 1012 – Specify TOC.

Reviewer #2: Overall, the work was very well designed, clearly written, and presents a novelty regarding the mechanism of action of a promising drug against leishmaniasis. Considering some suggested spelling corrections and the need to reinforce the issue of possible drug selectivity in the treatment of infection in mammals, which reinforces the importance of this work, it is able to be accepted for publication in PLOSNTD.

Reviewer #3: cTXNPx null mutant - I found the way that the diagnostic PCRs were shown misleading, and did not understand well the strategy. The final cell line contains 1 copy of LmxM.15.1040 deleted by BSDr gene, and 1 copy of LmxM.15.1160 by PACr gene? But how did the second copies of both genes were deleted by only NEOr in one go? Please clarify the strategy in the text and include in Fig 6A the agarose gels that show presence/absence of both CDS in each step of the genetic manipulation for better understanding.

PLOS authors have the option to publish the peer review history of their article (what does this mean?). If published, this will include your full peer review and any attached files.

Reviewer #1: No

Reviewer #2: No

Reviewer #3: No
---

## [Decision Letter · Decision Letter 1]

26 Oct 2021

Dear Prof Steel,

We are pleased to inform you that your manuscript 'Chalcones Identify cTXNPx as a Potential Antileishmanial Drug Target' has been provisionally accepted for publication in PLOS Neglected Tropical Diseases.

Best regards,

Tiago Donatelli Serafim

Associate Editor

Laura-Isobel McCall

Deputy Editor

Reviewer's Responses to Questions

**Key Review Criteria Required for Acceptance?**

**Methods**

-Are the objectives of the study clearly articulated with a clear testable hypothesis stated?

-Is the study design appropriate to address the stated objectives?

-Is the population clearly described and appropriate for the hypothesis being tested?

-Is the sample size sufficient to ensure adequate power to address the hypothesis being tested?

-Were correct statistical analysis used to support conclusions?

-Are there concerns about ethical or regulatory requirements being met?

Reviewer #3: The authors provided the information requested during revision.

**Results**

-Does the analysis presented match the analysis plan?

-Are the results clearly and completely presented?

-Are the figures (Tables, Images) of sufficient quality for clarity?

Reviewer #3: I am satisfied with the quality of figures and the description of the results, all of which match what is presented by the authors.

**Conclusions**

-Are the conclusions supported by the data presented?

-Are the limitations of analysis clearly described?

-Do the authors discuss how these data can be helpful to advance our understanding of the topic under study?

-Is public health relevance addressed?

Reviewer #3: The authors made the appropriate corrections in text and figures and addressed the concerns. Text is correct and addresses the advances to the field and public health relevance of the work.

**Editorial and Data Presentation Modifications?**

Reviewer #3: No modifications needed.

**Summary and General Comments**

Reviewer #3: No comments.

PLOS authors have the option to publish the peer review history of their article (what does this mean?). If published, this will include your full peer review and any attached files.

Reviewer #3: No

---

## [Editor Report · Acceptance letter]

7 Nov 2021

Dear Prof Steel,

We are delighted to inform you that your manuscript, "Chalcones Identify cTXNPx as a Potential Antileishmanial Drug Target," has been formally accepted for publication in PLOS Neglected Tropical Diseases.

Best regards,

Shaden Kamhawi

co-Editor-in-Chief

Paul Brindley

co-Editor-in-Chief
